# Synthesis of Novel Lipophilic Polyamines via Ugi Reaction and Evaluation of Their Anticancer Activity

**DOI:** 10.3390/molecules27196218

**Published:** 2022-09-21

**Authors:** Artemiy Nichugovskiy, Varvara Maksimova, Ekaterina Trapeznikova, Elizaveta Eshtukova-Shcheglova, Igor Ivanov, Marianna Yakubovskaya, Kirill Kirsanov, Dmitry Cheshkov, Gian Cesare Tron, Mikhail Maslov

**Affiliations:** 1Lomonosov Institute of Fine Chemical Technologies, MIREA—Russian Technological University, 86 Vernadsky Ave., 119571 Moscow, Russia; 2N.N. Blokhin National Medical Research Center of Oncology, 23 Kashirskoe Sh., 115478 Moscow, Russia; 3I.M. Sechenov First Moscow State Medical University, 8-2 Trubetskaya Str., 119991 Moscow, Russia; 4Institute of Medicine, Peoples’ Friendship University of Russia, 6 Miklukho-Maklaya Str., 117198 Moscow, Russia; 5State Scientific Research Institute of Chemistry and Technology of Organoelement Compounds, 38 Shosse Entuziastov, 105118 Moscow, Russia; 6Dipartimento di Scienza del Farmaco, Università del Piemonte Orientale, 2 Largo Donegani, 28100 Novara, Italy

**Keywords:** polyamines, multicomponent Ugi reaction, lipophilic polyamines, anticancer activity

## Abstract

Natural polyamines (PAs) are involved in the processes of proliferation and differentiation of cancer cells. Lipophilic synthetic polyamines (LPAs) induce the cell death of various cancer cell lines. In the current paper, we have demonstrated a new method for synthesis of LPAs via the multicomponent Ugi reaction and subsequent reduction of amide groups by PhSiH_3_. The anticancer activity of the obtained compounds was evaluated in the A-549, MCF7, and HCT116 cancer cell lines. For the first time, it was shown that the anticancer activity of LPAs with piperazine fragments is comparable with that of aliphatic LPAs. The presence of a diglyceride fragment in the structure of LPAs appears to be a key factor for the manifestation of high anticancer activity. The findings of the study strongly support further research in the field of LPAs and their derivatives.

## 1. Introduction

According to the latest statistics, about 19.3 million cancer cases and 10 million cancer-associated deaths are annually reported worldwide [1]. Currently, the search for new chemotherapeutic agents inhibiting invasion and metastasis faces the problem of resistance of cancer cells due to their somatic changes [2,3]. In this regard, modern biomedical approaches require new therapeutic strategies and development of anticancer agents to overcome these challenges.

Natural polyamines (PAs) putrescine, spermidine, and spermine that are present in significant amounts in all eukaryotic cells are essential for various underlying cellular processes such as proliferation, differentiation, and apoptosis [4]. They are formed inside the cell but can also be obtained from exogenous sources. Exogenous PAs penetrate into the cell by active transport and, once inside, are distributed in all cellular compartments due to their high solubility [5]. In eukaryotic cells, the intracellular concentration of PAs is strictly controlled by the mechanisms of their biosynthesis, catabolism, transport, and excretion. Uptake and biosynthesis of PAs grows up in response to proliferation stimuli. At the same time, catabolism and secretion of PAs, as well as inhibition of their biosynthesis and transport, are induced when higher PA concentrations are reached in the cell [6]. The levels of PAs in cancer cells are higher than in normal cells, and this phenomenon is associated with a high rate of cell proliferation, decreased level of apoptosis, and overexpression of genes that affect cancer invasion and metastasis [7].

The first synthesis of the norspermine derivatives **1**, **2** (Figure 1) that inhibit the growth of cancer cells was carried out in 1993 [8]. At present, dozens of PA derivatives with potential anticancer activity have been developed [9]. Although some of them (**3**–**5**) have been tested at different stages of clinical trials, none of them have been approved so far for medical use due to their low selectivity against cancer cells [4]. The lack of selectivity of anticancer agents based on PA structures stimulates further search for novel PA derivatives with improved properties for potential chemotherapeutic application.

Most of the known approaches regarding the synthesis of PA derivatives and their conjugates have several disadvantages [9], namely: (1) multistage synthetic procedures, (2) the introduction of orthogonal protective groups to block internal and terminal nitrogen atoms, (3) the low overall yield of the desired molecules, and (4) complicated purification procedures that are required for highly polar compounds. Following this approach, a synthetic scheme for the preparation of a family of PAs (**6**) containing an alkyl diglyceride fragment and an ethyl residue attached to terminal nitrogen atoms was developed in our laboratory [10]. Within this structure, the long-chain alkyl substituent (C_10_–C_18_) was placed at the C(1) atom of glycerol, whereas the short-chain ethyl substituent was placed at C(2). The presence of an alkyl group at the terminal nitrogen atom of polyamine slightly increases cytotoxicity compared to analogues with a free NH_2_ terminal group. This effect may be related to the fact that the terminal alkyl group prevents potential acylation and further oxidation of the compound, which increases its stability in cells [11]. The key step of the synthesis regarded the interaction of alkyl diglyceride bromides with regioselectively protected PAs under Fukuyama reaction conditions [12]. The low yields of compound **6** and the multistage nature of the synthetic scheme revealed the disadvantages of the proposed method.

The lipophilic PA may effectively inhibit PA transport into the cell due to its effective incorporation into the transmembrane channel located on the cell membrane. These data have been previously reported for AMXT-1501 with the palmitic acid residue [13]. In addition, we have previously shown that lipophilic PAs, where the lipophilic part is presented by a diglyceride fragment, also exhibit high anticancer activity [10]. Considering the results of the mentioned above studies, conjugation of PAs with the diglyceride fragment may have beneficial pharmacological potential.

The multicomponent Ugi reaction [14] can be used as an effective tool for the rapid preparation of modified PAs. One of the modifications of this reaction (*N*-split-Ugi [15,16]) is based on the interaction of a secondary diamine, a carbonyl compound, a carboxylic acid, and an isocyanide, which together form an α-acylaminoamide, whose amide groups can be further reduced to form a PA. This modification makes it possible to obtain PAs of different structures in two steps from simple compounds [17].

In this work, we implemented the multicomponent *N*-split Ugi reaction for the synthesis of novel alkylated PAs containing aliphatic and cyclic diamines and evaluated their anticancer activity.

## 2. Results and Discussion

The synthesis of lipophilic PAs using the *N*-split Ugi reaction is usually carried out in two steps. On the first step, α-acylaminoamide is formed by the condensation of four components. On the second step, the reduction of amide groups is carried out followed by the removal of protective groups. In this work, the commercially available *tert*-butyl isocyanide (**7a**) or the previously obtained octadecyl isocyanide (**7b**) [18] were used as isonitrile components, glacial acetic acid (**8a**) or *N*-acetylglycine (**8b**) as the carboxyl component, and *N*,*N*’-dibenzylalkanediamine (**9a**–**c**) as the diamine component, while paraformaldehyde (**10**) was used as the carbonyl component. The reaction was refluxed in methanol in an equimolar ratio of starting reagents for 16 h (Figure 1). Usually the N-split Ugi reaction proceeds under room conditions, but we used refluxing to dissolve the lipophilic isocyanide and to break paraformaldehyde completely.

A noticeably increased yield of the *N*-split Ugi reaction from 10–11% to 35–48% was observed when the length of the methylene linker between the central nitrogen atoms of diamines was increased from two (compounds **11a**,**b**) to three carbon atoms (compounds **11c**,**d**). On the contrary, further increase in its length to one additional methylene group decreases the yield of α-acylaminoamide **11e** to 30%. The obtained yields correlated with the previously published data [15]. In the NMR spectra of the α-acylaminoamides **11a**–**e**, appearance of double sets of signals which correlate with the formation of rotamers around the amide bonds was detected [19].

Piperazine is one of the widely used structural fragments in numerous biologically active compounds. Various piperazine derivatives demonstrated a high antiproliferative activity against different cancer cell lines [20,21,22,23,24]. The replacement of aliphatic diamines with piperazine results in increased conformational rigidity and lipophilicity, altering the proteolytic [25,26] and biological activity of PAs. Compounds **12a**–**e** with a piperazine fragment were obtained as described above for compounds **11a**–**e**. The replacement of aliphatic diamines **9a**–**c** with piperazine (**9d**) increases the yields of α-acylaminoamides **12a**–**e** and reduces the reaction time from 16 to 12 h (Figure 2).

The highest yield (compound **12b**) was achieved using octadecyl isocyanide (**7b**) and acetic acid (**8a**). Although the four-component *N*-split-Ugi reaction seems to be insensitive to steric hindrances [27], the yields of compounds **12c**,**d** obtained from the diglyceride **7c** were significantly lower, suggesting that steric hindrance caused by the ethyl substituent at the C(2) atom of glycerol might be the reason for the observed effect. Additionally, the low yield of compounds **12c,d** can be linked with lower stability of isocyanide **7c** and its partial transformation into formamide, as evidenced by the presence of the corresponding spot on TLC and NMR data of isolated formamide. The use of *N*-acetylglycine (**9b**) as a carboxyl component resulted in a decreased yield of α-acylaminoamides **12a**,**c**, which may be due to reduced nucleophilicity of the carbonyl carbon atom that undergoes the Mumm rearrangement [28].

In the ^13^C NMR spectra of compounds **12c**,**d**, no signal of carbonyl carbon at the diglyceride fragment was observed when CDCl_3_ was used as a solvent. At the same time, a strongly broadened signal of the corresponding NH-proton was detected in the ^1^H NMR spectra. Apparently, the intermolecular exchange of amide protons led to the strong broadening of the ^13^C signal of the corresponding carbonyl atom and its merging with the base line of the spectrum. To avoid those problems, the spectra of compounds **12c,d** were recorded in DMSO-d6, a solvent that somewhat suppresses the exchange of mobile protons.

Since cancer cells generally overexpress carbohydrate receptors, we attempted to prepare PAs that contain a diglyceride moiety at one terminal nitrogen atom and a carbohydrate moiety at the other via a two-step strategy using 2-(hydroxymethyl)benzoic acid [29]. This approach allows to prepare aminoamide, which acts as one of the components in the *N*-split Ugi reaction. Isonitrile **7c** reacted with equimolar amounts of 2-(hydroxymethyl)benzoic acid (**8c**), piperazine (**9d**), and formaldehyde (**10**) (Figure 3) to give monosubstituted aminoamide **13a** in a 35% yield. The low yield of the desired product **13a** was due to the formation of the symmetrical adduct of aminodiamide **13b** with 22% yield. Subsequent treatment of **13a** with isocyanide **14a**–**c**, 2-(hydroxymethyl)benzoic acid (**8c**), and formaldehyde (**10**) led to formation of the disubstituted piperazines **15a**–**c** in 75%, 80%, and 30% yields, respectively. The low yield of D-glucose containing compound **15c** is supposedly associated with a partial deacetylation during reflux.

The most common method to reduce the amide group to the corresponding amine utilizes LiAlH_4_ [30] or BH_3_ and its derivatives [31,32]. Unfortunately, in the case of our compounds, stable boron-amine complexes were formed which could not be further hydrolyzed to the desired amines **15a**–**f** under basic or acidic conditions. Therefore, we applied phenylsilane and NiCl_2_(dme) [33] for a chemoselective aminoamides reduction, using 2 equivalents of phenylsilane and 0.1 equivalent of NiCl_2_(dme) per each amide group. As reported previously [34], the utilization of benzamide-type substrates is one of the key limitations for the most nickel-catalyzed amide reduction reactions. Indeed, using the abovementioned strategy, benzylamide-derivative **16a** was obtained in a good yield of 66%, whereas the yields of piperazyl derivatives **16b**–**f** were significantly lower (Figure 4).

Treatment of carbohydrate containing aminoamide **1****5c** with phenylsilane did not provide the formation of the desired amine (Figure 5) due to partial deacetylation of the D-glucose. To overcome this problem, the acetyl groups of aminoamide **15c** were initially removed by sodium methoxide in methanol to yield compound **17** (80%). The following reduction of the amide **17** was unsuccessful, and the desired amine was not isolated from the reaction mixture. Thus, the reduction of the amide groups of aminodiamide **15c** containing D-glucose requires additional efforts to find alternative synthetic approaches.

Given the fact that Pas with a piperazyl domain have never been reported before, we chose piperazyl derivatives **16b** and **16c**–**f** with different hydrophobic domain structures; with short-chain substituents (ethyl (**16c**), isopropyl (**16e**), and pentyl (**16f**)) and different numbers of amino groups. To evaluate the effect of lipophilic PA structure on its anticancer activity, other aliphatic lipophilic PAs, which were obtained in this study in much lower yields, were not considered for further evaluation. Aminoamide **17** was used as the negative control.

The cytotoxicity of the lipophilic PAs **16a**–**f** was determined using the MTT-test (see Appendix A) in breast cancer (MCF7), human lung adenocarcinoma (A549), colon cancer (HCT116) cell lines (Table 1). The cytotoxicity data showed that the presence of a diglyceride fragment as a hydrophobic domain (PAs **16c**–**f**) increases their anticancer activity compared with the octadecyl substituent (PA **16b**).

Compound **16d** with three amino groups showed the highest anticancer activity within all cell lines tested. Compounds with four amino groups **16c**,**e**,**f** revealed similar anticancer activity. BENSpm (**3**) and the widely used anticancer agent cisplatin were selected as a positive control. Their IC_50_ values obtained in this study were close to or of the same value as those obtained in previous reports [35,36,37]. Comparison of IC_50_ values obtained suggests that new lipophilic PAs **16c**–**f** have a cytotoxicity that is comparable to that of BENSpm, and several times higher than that of cisplatin.

## 3. Conclusions

Lipophilic Pas manifest excellent preliminary biological activity in cancer cell lines. However, the chemical synthesis of such compounds is complicated. In this paper, we demonstrated the efficient approach for the synthesis of new LPAs, which were obtained using the *N*-split Ugi multicomponent reaction. The application of this method allowed us to decrease the synthetic steps and to increase the total yield of LPAs from 7% to 28%. The application of PhSiH_3_ and NiCl_2_(dme) effectively permitted us to reduce several amide groups in the PA precursors and has proven to be a very reliable and efficient method.

The obtained results demonstrate that the biological activity of the novel LPAs is several times higher than that of cisplatin, which is used in medical practice. At the same time, comparison with the clinically tested BENSpm showed similar cytotoxicity, which makes LPAs promising targets for further studies. More detailed biological evaluation will be carried out in the follow up study.

## 4. Materials and Methods

### 4.1. General

Commercially available solvents were used in this study. All the experiments were carried out under argon atmosphere with the use of the HPLC grade methanol. The reactions were monitored by thin-layer chromatography (TLC) on Silica gel 60 F_254_ plates (Merck, Germany). The substances were identified in UV light (254 nm) by the treatment with Dragendorff’s reagent, or by treatment with a solution of phosphomolybdic acid–cerium sulfate (IV) with subsequent heating. Column chromatography was performed on Kieselgel 60 silica gel (0.040–0.063 or 0.063–0.200 mm, Merck, Germany). The ^1^H and ^13^C NMR spectra were recorded on Bruker DPX-300, Bruker Avance II 400, or Bruker Avance II 600 Fourier spectrometers (Bruker, Germany) in CDCl_3_, DMSO-*d*6 or acetone-*d*6. Chemical shifts (δ) were expressed in ppm relative to the peak of the residual proton of the solvent. The spin–spin interaction constants (*J*) are reported in Hz. The high-resolution mass spectra were recorded on a LCQ Deca XP Plus mass spectrometer with ESI ionization (Thermo Finnigan, San Jose, CA, USA) or FT ICR Apex Ultra 7 T (Bruker, Germany), mass spectra were recorded on the Agilent spectrometer. LCMS spectra were recorded on the LC Agilent Infinity 1260 II (Agilent, Beijing, China) and MSD Agilent IQ (Agilent, Singapore). Column PoroShell 120 EC-C18, 100 mm × 4.6 mm × 3 µm, constant flow 800 µL/min, linear gradient from 90% water + 0.1% FA—0–2 min to 90% ACN + 0.1% FA—15–25 min, voltage of ion capillary 3500 V, fragmenter 100 V.

2-Hydroxymethylbenzoic acid (**8c**) was prepared as described previously [29]. The synthesis of isonitrile derivatives both of diglyceride (**7c**) and D-glucose was performed according to [18]. The synthesis of PhSiH_3_ was described in reference [38].

To eliminate minor impurities compounds **16b**–**f** that have been evaluated in cell models were additionally purified prior to use on silica gel and their purity (≥96%) was confirmed by LCMS method.

### 4.2. Synthetic Methods

#### 4.2.1. General Procedure for the Synthesis for Compounds **11a**–**e**, **12a**–**e**

Isocyanide **7** (1 eq), carboxylic acid **8** (1 eq), and diamine **9** (1 eq) were added sequentially to a solution of paraformaldehyde **10** (1 eq) in methanol (0.5 M). The reaction mixture was at refluxed for 12–16 h. The solvent was evaporated, and the crude reaction mixture was purified by column chromatography.

##### 1,8-Diamino-*N*^8^-Acetyl-*N*^1^-Tert-Butyl-1,7-Dioxo-*N*^3^,*N*^6^-Dibenzyl-3,6-Diazaoctane (**11a**)

Yield: 11%, colorless oil. Eluent: CHCl_3_-MeOH (10:1). ^1^H NMR (600 MHz, acetone-d6, main rotamer) δ 1.31 (s, 9H, (CH_3_)_3_), 1.93 (s, 3H, COCH_3_), 2.62 (br. s, 2H, CH_2_NCH_2_), 3.05 (s, 2H, COCH_2_N), 3.50 (s, 2H, CH_2_NCO), 3.60 (s, 2H, COCH_2_NH), 4.05 (d, 2H, *J* = 4.8 Hz, PhCH_2_), 4.37 (s, 2H, PhCH_2_NCO), 7.07–7.46 (m, 12H, 2 Ph, 2 NH). ^13^C NMR (150 MHz, acetone-d6, main rotamer) δ 22.7, 29.0, 42.0, 44.4, 50.4, 52.1, 59.5, 60.0, 127.6, 128.3, 128.7, 129.4, 129.4, 129.7, 137.7, 138.7, 169.6, 170.2, 170.4. HRMS ESI *m*/*z*: [M + H]^+^ calcd for C_26_H_37_N_4_O_3_ 453.2860, found: 453.2860.

##### 1,8-Diamino-*N*^8^-Acetyl-*N*^1^-Octadecyl-1,7-Dioxo-*N*^3^,*N*^6^-Dibenzyl-3,6-Diazaoctane (**11b**)

Yield: 10%, colorless oil. Eluent: CHCl_3_-MeOH (10:1). ^1^H NMR (300 MHz, CDCl_3_, main rotamer) δ 0.88 (t, 3H, *J* = 7.0 Hz, (CH_2_)_15_CH_3_), 1.25 (br. s, 30H, (CH_2_)_15_CH_3_), 1.39–1.57 (m, 2H, CH_2_CH_2_(CH_2_)_15_), 2.05 (s, 3H, COCH_3_), 2.59 (t, 2H, *J* = 6.5 Hz, COCH_2_NCH_2_), 3.16 (s, 2H, COCH_2_N), 3.20–3.35 (m, 2H, CH_2_CH_2_(CH_2_)_15_), 3.47 (t, 2H, *J* = 6.5 Hz, CONCH_2_), 3.57 (s, 2H, COCH_2_NH), 4.04 (s, 2H, PhCH_2_), 4.14 (s, 2H, PhCH_2_NCO), 6.51 (s, 1H, NHCOCH_2_), 6.97–7.50 (m, 11H, 2 Ph, CH_3_CONH). ^13^C NMR (75 MHz, CDCl_3_, main rotamer) δ 14.1, 22.7, 29.3, 29.4, 29.6, 29.7, 29.7, 29.8, 31.9, 39.1, 41.6, 43.9, 49.6, 51.2, 59.0, 59.8, 125.3, 126.3, 128.2, 128.6, 129.0, 129.1, 135.0, 138.1, 168.8, 169.9, 170.6. HRMS ESI *m*/*z*: [M + H]^+^ calcd for C_40_H_65_N_4_O_3_ 649.5051, found: 649.5051.

##### 1,9-Diamino-*N*^9^-Acetyl-*N*^1^-Octadecyl-1,8-Dioxo-*N*^3^,*N*^7^-Dibenzyl-3,7-Diazanonane (**11c**)

Yield: 48%, colorless oil. Eluent: EA-MeOH (9:1). ^1^H NMR (400 MHz, CDCl_3_, COSY, HSQC, HMBC) δ 0.87 (t, 3H, *J* = 6.9 Hz, (CH_2_)_15_CH_3_), 1.25 (br. s, 30H, (CH_2_)_15_CH_3_), 1.38–1.52 (m, 2H, CH_2_CH_2_(CH_2_)_15_), 1.64–1.80 (m, 2H, NCH_2_CH_2_CH_2_N), 2.04 (s, 3H, COCH_3_), 2.36–2.58 (m, 2H, PhCH_2_NCH_2_), 3.06 (s, 2H, COCH_2_N), 3.09–3.31 (m, 3H, CH_2_CH_2_(CH_2_)_15_, PhCH_2_N(CO)CH_2_), 3.39 (t, 1H, *J* = 7.3 Hz, PhCH_2_N(CO)CH_2_), 3.57 (s, 2H, COCH_2_NH), 4.05 (d, 2H, *J* = 3.9 Hz, PhCH_2_), 4.41 (s, 2H, PhCH_2_NCO), 6.57 (s, 1H, NHCOCH_2_), 6.84–7.54 (m, 11H, 2 Ph, CH_3_CONH). ^13^C NMR (101 MHz, CDCl_3_) δ 14.1, 22.7, 23.0, 27.0, 29.3, 29.3, 29.6, 29.6, 29.6, 29.7, 29.7, 31.9, 39.0, 41.5, 44.2, 48.8, 50.1, 52.0, 58.0, 59.9, 126.3, 127.9, 128.1, 128.7, 128.8, 129.1, 135.3, 136.6, 168.1, 170.0, 170.1. HRMS ESI *m*/*z*: [M + H]^+^ calcd for C_41_H_67_N_4_O_3_ 663.5208, found: 663.5196. HRMS ESI *m*/*z*: [M + Na]^+^ calcd for C_41_H_66_NaN_4_O_3_ 685.5033, found: 685.5001.

##### 1,6-Diamino-*N*^1^-Acetyl-*N*^6^-Octadecyl-6-Oxo-*N*^1^,*N*^3^-Dibenzyl-4-Azahexane (**11d**)

Yield: 35%, colorless oil. Eluent: PE-EA (4:6). ^1^H NMR (400 MHz, CDCl_3_) δ 0.80 (t, 3H, *J* = 6.6 Hz, (CH_2_)_15_CH_3_), 1.18 (br. s, 30H, (CH_2_)_15_CH_3_), 1.29–1.41 (m, 2H, CH_2_CH_2_(CH_2_)_15_), 1.55–1.70 (m, 2H, NCH_2_CH_2_CH_2_N), 2.01 (s, 3H, COCH_3_), 2.31–2.43 (m, 2H, PhCH_2_NCH_2_), 2.96 (s, 2H, COCH_2_N), 3.05–3.17 (m, 2H, CH_2_CH_2_(CH_2_)_15_), 3.29 (t, 2H, *J* = 7.5 Hz, PhCH_2_N(CO)CH_2_), 3.48 (s, 2H, PhCH_2_), 4.37 (s, 2H, PhCH_2_NCO), 6.76–7.38 (m, 11H, 2 Ph, NH). ^13^C NMR (101 MHz, CDCl_3_) δ 14.0, 21.3, 21.7, 22.6, 25.1, 26.1, 26.9, 29.2, 29.2, 29.2, 29.5, 29.5, 29.5, 29.6, 31.8, 38.8, 38.9, 43.6, 46.0, 48.2, 51.9, 52.1, 52.3, 57.9, 59.6, 126.1, 127.3, 127.3, 127.5, 127.6, 127.6, 127.9, 128.2, 128.4, 128.5, 128.5, 128.6, 128.6, 128.7, 128.7, 128.8, 129.0, 129.5. HRMS ESI *m*/*z*: [M + H]^+^ calcd for C_39_H_63_N_3_O_2_ 606.4993, found: 606.4991.

##### 1,7-Diamino-*N*^1^-Acetyl-*N*^7^-Octadecyl-7-Oxo-*N*^1^,*N*^4^-Dibenzyl-5-Azaheptane (**11e**)

Yield: 30%, colorless oil. Eluent: EA. ^1^H NMR (300 MHz, main rotamer, CDCl_3_) δ 0.71 (t, 3H, *J* = 6.7 Hz, (CH_2_)_15_CH_3_), 1.12 (br.s, 30H, (CH_2_)_15_CH_3_,), 1.19–1.41 (m, 6H, CH_2_CH_2_(CH_2_)_15_, NCH_2_(CH_2_)_2_CH_2_N), 1.93 (s, 3H, COCH_3_), 2.21–2.36 (m, 2H PhCH_2_NCH_2_,), 2.88 (s, 2H, NCH_2_CO), 2.94–3.22 (m, 4H, COCH_2_N, CH_2_CH_2_(CH_2_)_15_), 3.40 (s, 2H, PhCH_2_), 4.32 (s, 2H, PhCH_2_NCO), 6.78–7.31 (m, 2 Ph, NH, 11H). ^13^C NMR (75 MHz, CDCl_3_) δ 14.2, 21.6, 21.9, 22.8, 24.7, 25.3, 26.4, 27.1, 29.4, 29.8, 32.0, 39.0, 46.0, 47.9, 48.3, 52.2, 54.8, 58.1, 59.7, 59.9, 126.3, 127.5, 127.7, 128.0, 128.6, 128.8, 128.9, 129.0, 136.9, 137.7, 138.1, 138.2, 170.9, 171.2. HRMS FTICR *m*/*z*: [M + H]^+^ calcd for C_40_H_66_N_3_O_2_ 620.5150, found: 620.5136.

##### *N*^1^-(*N*-Acetylglycyl)-*N*^4^-[(*N*-Octadecyl)Aminocarbonyl]Methylpiperazin (**12a**)

Yield: 60%, colorless oil. Eluent: DCM-MeOH (20:1). ^1^H NMR (300 MHz, CDCl_3_, ^1^H-^1^H COSY) δ 0.84 (d, 3H, *J* = 6.9 Hz (CH_2_)_15_CH_3_), 1.22 (br. s, 30H, (CH_2_)_15_CH_3_), 1.41–1.54 (m, 2H, CH_2_CH_2_(CH_2_)_15_), 2.01 (s, 3H, COCH_3_), 2.47–2.55 (m, 4H, 2 COCH_2_NCH_2_ Pip), 3.00 (s, 2H, COCH_2_N), 3.19–3.29 (m, 2H, CH_2_CH_2_(CH_2_)_15_), 3.38–3.46 (m, 2H, 2 CONCH_e_H_a_ Pip), 3.60–3.66 (m, 2H, 2 CONCH_e_H_a_ Pip), 4.02 (d, 2H, *J* = 4.1 Hz, COCH_2_NH), 6.61 (t, 1H, *J* = 4.1 Hz, NHCOCH_3_), 6.92 (t, 1H, *J* = 5.5 Hz, CH_2_CONH). ^13^C NMR (75 MHz, CDCl_3_) δ 14.1, 22.6, 22.9, 27.0, 29.2, 29.3, 29.5, 29.6, 29.6, 29.7, 29.7, 31.9, 39.0, 41.2, 42.0, 44.4, 53.0, 53.2, 61.5, 166.6, 169.0, 170.1. HRMS FTICR *m*/*z*: [M + H]^+^ calcd for C_28_H_55_N_4_O_3_ 495.4269, found: 495.4269.

##### *N*^1^-Acetyl-*N*^4^-[(N-Octadecyl)Aminocarbonyl]Methylpiperazin (**12b**)

Yield: 80%, colorless oil. Eluent: DCM-MeOH (30:1). ^1^H NMR (300 MHz, CDCl_3_, ^1^H-^1^H COSY) δ 0.90 (t, 3H, *J* = 7.0 Hz, (CH_2_)_15_CH_3_), 1.26 (br. s, 30H, (CH_2_)_15_CH_3_), 1.48–1.61 (m, 2H, CH_2_CH_2_(CH_2_)_15_), 2.11 (s, 3H, COCH_3_), 2.49–2.59 (m, 4H, 2 COCH_2_NCH_2_ Pip), 3.05 (s, 2H, COCH_2_N), 3.20–3.37 (m, 2H, CH_2_CH_2_(CH_2_)_15_), 3.46–3.54 (m, 2H, 2 CONCH_e_H_a_ Pip), 3.63–3.69 (m, 2H, 2 CONCH_e_H_a_ Pip), 7.06 (br. s, 1H, NH). ^13^C NMR (75 MHz, CDCl_3_) δ 14.1, 21.3, 22.7, 26.9, 29.2, 29.3, 29.5, 29.6, 29.6, 29.7, 31.9, 39.0, 41.3, 46.2, 53.1, 53.5, 61.5, 169.0, 169.1. HRMS FTICR *m*/*z*: [M + H]^+^ calcd for C_28_H_55_N_4_O_3_ 438.4054, found: 438.4054.

##### *N*^1^-(*N*-Acetylglycyl)-*N*^4^-[*N*-(*rac*-1-Decyloxy-2-Ethyloxyprop-3-yl)Aminocarbonyl]Methylpiperazin (**12c**)

Yield: 47%, colorless oil. Eluent: DCM-MeOH (15:1). ^1^H NMR (600 MHz, DMSO-*d6*, COSY, HSQC, HMBC) δ 0.85 (t, 3H, *J* = 6.9 Hz, (CH_2_)_7_CH_3_), 1.09 (t, 3H, *J* = 7.0 Hz, OCH_2_CH_3_), 1.24 (br. s, 14H, (CH_2_)_7_CH_3_), 1.43–1.51 (m, 2H, OCH_2_CH_2_), 1.86 (s, 3H, COCH_3_), 2.34–2.42 (m, 2H, 2 COCH_2_NCH_e_H_a_ Pip), 2.42–2.48 (m, 2H, 2 COCH_2_NCH_e_H_a_ Pip), 2.93 (d, *J* = 15.5 Hz, 1H, COCH_a_H_b_N), 2.96 (d, *J* = 15.5 Hz, 1H, COCH_a_H_b_N), 3.07–3.13 (m, 1H, CONHCH_a_H_b_), 3.25–3.32 (m, 1H, NHCH_a_H_b_CH), 3.33–3.40 (m, 4H, CH_2_OCH_2_), 3.40–3.52 (m, 6H, CHOCH_a_H_b_CH_3_, 2 CONCH_2_ Pip), 3.52–3.59 (m, 1H, OCH_a_H_b_CH_3_), 3.92 (d, 2H, *J* = 5.5 Hz, COCH_2_NH), 7.63 -7.70 (m, 1H, CHCH_2_NH), 7.91 (t, 1H, *J* = 5.5 Hz, COCH_2_NH). ^13^C NMR (151 MHz, DMSO-*d6*) δ 13.9, 15.5, 22.0, 22.4, 25.6, 28.7, 28.8, 29.0, 29.0, 29.0, 29.0, 29.1, 31.3, 39.6, 40.3, 41.3, 44.0, 52.4, 52.7, 60.9, 64.4, 70.6, 71.1, 76.3, 167.0, 168.8, 169.2. HRMS FTICR *m*/*z*: [M + H]^+^ calcd for C_27_H_53_N_4_O_5_ 513.4010, found: 513.4010.

##### *N*^1^-Acetyl-*N*^4^-[*N*-(*rac*-1-Decyloxy-2-Ethyloxyprop-3-yl)Aminocarbonyl]Methylpiperazin (**12d**)

Yield: 54%, colorless oil. Eluent: DCM-MeOH (30:1). ^1^H NMR (600 MHz, DMSO-*d*_6_, HSQC, HMBC) δ 0.85 (t, 3H, *J* = 7.0 Hz, (CH_2_)_7_CH_3_), 1.09 (t, 3H, *J* = 7.0 Hz, OCH_2_CH_3_), 1.24 (br. s, 14H, (CH_2_)_7_CH_3_), 1.44–1.51 (m, 2H, OCH_2_CH_2_), 1.98 (s, 3H, COCH_3_), 2.35–2.39 (m, 2H, 2 COCH_2_NCH_e_H_a_ Pip), 2.43–2.45 (m, 2H, 2 COCH_2_NCH_e_H_a_ Pip), 2.93 (d, 1H, *J* = 15.5 Hz, COCH_a_H_b_N), 2.96 (d, 1H, *J* = 15.5 Hz, COCH_a_H_b_N), 3.05–3.13 (m, 1H, CONHCH_a_H_b_), 3.29 (ddd, 1H, *J* = 13.4, 6.3, 5.1 Hz, CONHCH_a_H_b_), 3.32–3.39 (m, 4H, CH_2_OCH_2_), 3.40–3.52 (m, 6H, CHOCH_a_H_b_CH_3_, 2 CONCH_2_ Pip), 3.52–3.59 (m, 1H, CHOCH_a_H_b_CH_3_), 7.63–7.68 (m, 1H, NH). ^13^C NMR (151 MHz, DMSO-*d*_6_, DEPT-135) δ 13.8, 15.5, 22.0, 25.6, 28.7, 28.8, 29.0, 29.0, 29.0, 29.0, 29.1, 31.3, 39.6, 40.7, 45.6, 52.4, 52.9, 60.9, 64.4, 64.4, 70.6, 71.1, 76.3, 168.0, 168.8. HRMS FTICR *m*/*z*: [M + H]^+^ calcd for C_28_H_55_N_4_O_3_ 456.3796, found: 456.3796.

##### *N*^1^-Acetyl-*N*^4^-[*N*-(Cyclohexyl)Aminocarbonyl]Methylpiperazin (**12e**)

Yield: 75%, colorless oil. Eluent: EA-MeOH (4:1). ^1^H NMR (300 MHz, CDCl_3_) δ 1.04–1.25 (m, 3H, 2 CHCH_2_H_e_H_a_, CHCH_2_CH_2_H_e_H_a_), 1.25–1.45 (m, 2H, 2 CHCH_2_CH_e_H_a_), 1.61 (m, 3H, 2 NHCHCH_e_H_a_, CHCH_2_CH_2_CH_e_H_a_), 1.77–1.90 (m, 2H, 2 NHCHCH_e_H_a_), 2.04 (s, 3H, COCH_3_), 2.35–2.57 (m, 4H, 2 COCH_2_NCH_2_ Pip), 2.96 (s, 2H, COCH_2_N), 3.39–3.50 (m, 2H, 2 CONCH_e_H_a_ Pip), 3.54–3.64 (m, 2H, 2 CONCH_e_H_a_ Pip), 3.65–3.86 (m, 1H, CONHCH), 6.88 (d, 1H, *J* = 8.2 Hz, NH). ^13^C NMR (75 MHz, CDCl_3_) δ 21.3, 24.7, 25.5, 33.1, 41.4, 46.3, 47.5, 53.1, 53.5, 61.6, 168.4, 169.0. HRMS ESI *m*/*z*: [M + H]^+^ calcd for C_14_H_26_N_3_O_2_ 268.20195, found: 268.20195.

##### 4.2.2. Synthesis of Compounds **13a,b**

The equimolar solution of isocyanide (**7c**), 2-(hydroxymethyl)benzoic acid (**8c**), amine (**9d**) and paraformaldehyde (**10**), and in 0.5 M methanol was refluxed for 12 h. The solvent was evaporated, and the crude reaction mixture was purified by column chromatography.

##### *N*^1^-[*N*-(*rac*-1-Decyloxy-2-Ethyloxyprop-3-yl)Aminocarbonyl]Methylpiperazin (**13a**)

Yield: 50%, colorless oil. Eluent: EA-MeOH-NH_3_·H_2_O (7:3:0.1) ^1^H NMR (300 MHz, CDCl_3_) δ 0.87 (t, *J* = 6.3 Hz, 3H, (CH_2_)_7_CH_3_), 1.21 (t, 3H, *J* = 7.0 Hz, OCH_2_CH_3_), 1.26 (br.s, 14H, (CH_2_)_7_CH_3_), 1.48–1.63 (m, 2H, OCH_2_CH_2_), 2.43–2.63 (m, 5H, CH_2_NHCH_2_ Pip), 2.87–2.96 (m, 4H, 2 NCH_2_ Pip), 2.99 (s, 2H, COCH_2_N), 3.17–3.75 (m, 9H, CH_2_OCH_2_, CHOCH_2_CH_3_, CH_2_NHCO), 7.50 (br.s, 1H, CONH). ^13^C NMR (75 MHz, CDCl_3_) δ 14.3, 15.8, 22.8, 26.2, 29.5, 29.6, 29.7, 29.8, 40.2, 46.1, 54.7, 62.3, 65.5, 71.5, 72.0, 76.9, 170.3. HRMS ESI [M + H]^+^ calcd for C_21_H_44_N_3_O_3_ 386.3377, found 386.3371.

##### *N*^1^,*N*^4^-bis[*N*-(*rac*-1-Decyloxy-2-Ethyloxyprop-3-yl)Aminocarbonyl]Methylpiperazin (**13b**)

Yield: 22%, colorless oil. Eluent: EA-MeOH (95:5). ^1^H NMR (300 MHz, CDCl_3_) δ 0.87 (t, 6H, *J* = 6.5 Hz, (CH_2_)_7_CH_3_), 1.19 (t, 6H, *J* = 7.0 Hz, OCH_2_CH_3_), 1.27 (br.s, 28H, (CH_2_)_7_CH_3_), 1.50–1.63 (m, 4H, OCH_2_CH_2_), 2.56 (br.s, 8H Pip), 3.01 (s, 4H, COCH_2_N), 3.16–3.31 (m, 2H, 2 CH_a_H_b_NHCO), 3.34–3.48 (m, 8H, 2 CHOCH_2_CH_3_, 2 CH_a_H_b_NHCO), 3.48–3.56 (m, 4H, 2 CH_2_OCH_2_), 3.56–3.74 (m, 4H, 2 CH_2_OCH_2_). ^13^C NMR (75 MHz, CDCl_3_) δ 14.3, 15.8, 22.8, 26.2, 29.5, 29.6, 29.7, 29.8, 29.8, 32.0, 40.3, 53.8, 61.7, 65.5, 71.5, 72.0, 76.8, 170.1. HRMS ESI [M + H]^+^calcd for C_38_H_77_N_4_O_6_ 685.5838, found 685.5828. HRMS ESI [M + 2H]^2+^calcd for C_38_H_78_N_4_O_6_ 343.2955, found 343.2954.

#### 4.2.3. General Procedure for the Synthesis of Compounds **15a**–**c**

The equimolar solution of corresponding isocyanide (**14a**–**c**), 2-(hydroxymethyl)benzoic acid (**8c**), amine (**13a**), and paraformaldehyde (**10**) in 0.5 M methanol was refluxed for 12 h. The solvent was evaporated, and the crude reaction mixture was purified by column chromatography.

##### *N*^1^-[*N*-(Isopropyl)Aminocarbonyl]Methyl-*N*^4^-[*N*-(*rac*-1-Decyloxy-2-Ethyloxyprop-3-yl)Aminocarbonyl]Methylpiperazin (**15a**)

Yield: 80%, colorless oil. Eluent: EA-MeOH (85:15). ^1^H NMR (300 MHz, CDCl_3_) δ 0.0.87 (t, *J* = 7.0 Hz, 3H, (CH_2_)_7_CH_3_), 1.16 (d, *J* = 6.6 Hz, 6H, CH(CH_3_)_2_), 1.20 (t, *J* = 7.0 Hz, 3H, OCH_2_CH_3_), 1.27 (br.s, 14H, (CH_2_)_7_CH_3_), 1.50–1.62 (m, 2H, OCH_2_CH_2_), 2.56 (br.s, 8H Pip protons), 2.97 (s, 2H, CHNHC(O)CH_2_), 3.03 (d, *J* = 1.6 Hz, 2H, COCH_2_N), 3.23 (ddd, *J* = 4.8, 6.4, 13.7 Hz, 1H, CHCH_a_H_b_NH), 3.34–3.75 (m, 8H, CHCH_a_H_b_NH, CHOCH_2_CH_3_, CH_2_OCH_2_), 4.01–4.16 (m, 1H, CH(CH_3_)_2_), 6.86 (br.d, *J* = 8.4 Hz, 1H, NHCH), 7.44 (br.t, *J* = 5.6 Hz, 1H, NHCH_2_). ^13^C NMR (75 MHz, CDCl_3_) δ 14.3, 15.8, 22.8, 23.0, 26.3, 29.5, 29.6, 29.7, 29.8, 29.8, 32.0, 40.3, 40.9, 53.7, 53.7, 61.7, 61.7, 65.5, 71.5, 72.0, 76.8, 169.0, 170.0. HRMS ESI [M + H]^+^ calcd for C_26_H_53_N_4_O_4_ 485.4061, found 485.4062.

##### *N*^1^-[*N*-(Pentyl)Aminocarbonyl]Methyl-*N*^4^-[*N*-(*rac*-(1-Decyloxy-2-Ethyloxyprop-3-yl)Aminocarbonyl]Methylpiperazin (**15b**)

Yield: 75%, colorless oil. Eluent: EA-MeOH (9:1). ^1^H NMR (400 MHz, CDCl_3_, COSY, HSQC, HMBC) δ 0.83–0.94 (m, 6H, (CH_2_)_7_CH_3_, (CH_2_)_4_CH_3_), 1.20 (t, 3H, *J* = 7.0 Hz, OCH_2_CH_3_), 1.23–1.39 (br. s, 18H, (CH_2_)_7_CH_3_, NHCH_2_CH_2_(CH_2_)_2_CH_3_), 1.46–1.63 (m, 4H, NHCH_2_CH_2_, OCH_2_CH_2_), 2.42–2.70 (br.s, 8H, Pip protons), 2.96–3.07 (m, 4H, 2 COCH_2_N), 3.18–3.32 (m, 3H, CHOCH_a_H_b_N, NHCH_2_CH_2_), 3.37–3.59 (m, 6H, 2 CH_2_OCH_2_, OCH_a_H_b_CH_3_, CHO), 3.59–3.76 (m, 2H, OCH_a_H_b_CH_3_, CHOCH_a_H_b_N), 7.09 (br. s, 1H, NH), 7.45 (br.s, 1H, NH). ^13^C NMR (101 MHz, CDCl_3_) δ 14.1, 14.2, 15.8, 22.4, 22.8, 26.2, 29.2, 29.4, 29.5, 29.6, 29.7, 29.7, 29.7, 32.0, 39.0, 40.2, 53.6, 53.7, 61.6, 61.6, 65.4, 71.5, 72.0, 76.8, 169.7, 170.0. MS ESI *m*/*z*: [M + H]^+^ calcd for C_28_H_57_N_4_O_4_ 513.44, found: 513.50.

##### *N*^1^-[*N*-(2,3,4,6-Tetra-*O*-Acetyl-β-D-Glucopyranosyl)Aminocarbonyl]Methyl-*N*^4^-[*N*-(*rac*-1-decyloxy-2-Ethyloxyprop-3-yl)Aminocarbonyl]Methylpiperazin (**15c**)

Yield: 30%, colorless oil. Eluent: EA-MeOH (95:5) ^1^H NMR (600 MHz, CDCl_3_) δ 0.85 (t, *J* = 7.0 Hz, 3H, (CH_2_)_7_CH_3_), 1.16 (t, 7.0 Hz, 3H, OCH_2_CH_3_), 1.21–1.31 (m, 14H, (CH_2_)_7_CH_3_), 1.49–1.57 (m, 2H, OCH_2_CH_2_), 1.96, 1.98, 2.0, 2.05 (s, 3H, 4 COCH_3_), 2.46 (m, 4H, CH_2_NCH_2_ Pip), 2.58 (br.s, 4H, CH_2_NCH_2_ Pip), 2.92 (dd, *J* = 2.3, 16.7 Hz, 1H, CHNHC(O)CH_a_H_b_), 3.02 (d, *J* = 16.4 Hz, 1H, CH_2_NHC(O)CH_a_H_b_), 3.05 (d, *J* = 16.4 Hz, 1H, CH_2_NHC(O)CH_a_H_b_), 3.09 (dd, *J* = 3.5, 16.7 Hz, 1H, CHNHC(O)CH_a_H_b_), 3.21 (dddd, *J* = 4.8, 6.8, 8.0, 13.8 Hz, 1H, CH_a_H_b_NH), 3.34–3.54 (m, 6H, CH_2_OCH_a_H_b_, CHOCH_2_CH_3_), 3.56–3.69 (m, 2H, CH_a_H_b_NH, CHCH_a_H_b_), 3.80 (ddd, *J* = 2.2, 4.4, 10.1 Hz, 1H, H-5), 4.05 (dd, *J* = 2.2, 12.5 Hz, 1H, H-6), 4.29 (dd, 1H, *J* = 4.4, 12.5 Hz, H-6), 4.98 (dd, *J* = 9.5, 9.6 Hz, H-2), 5.05 (dd, *J* = 9.4, 10.1 Hz, 1H, H-4), 5.22 (dd, *J* = 9.5, 9.8 Hz, 1H, H-1), 5.28 (dd, *J* = 9.4, 9.6 Hz, 1H, H-3), 7.81 (d, *J* = 9.8 Hz, 1H, CH_2_NH). ^13^C NMR (151 MHz, CDCl_3_) δ 14.2, 15.7, 20.6, 20.6, 20.6, 20.8, 22.7, 26.2, 29.4, 29.5, 29.6, 29.7, 29.7, 31.9, 40.2, 40.2, 53.2, 53.7, 61.4, 68.3, 70.5, 71.5, 71.5, 73.0, 73.8, 76.7, 76.7, 76.9, 77.2, 77.4, 77.8, 169.6, 169.9, 170.2, 170.2, 170.6, 171.1. HRMS ESI [M + H]^+^ calcd for C_37_H_65_N_4_O_13_ 773.4543, found 773.4535.

#### 4.2.4. General Procedure for Synthesis of Compounds **16a**–**f**

NiCl_2_(dme) (0.2 eq) and PhSiH_3_ (2 eq for each amide group) were added into a cylindrical pressure vessel with corresponding amide in toluene (1 M). The mixture was flushed with argon, tightly closed, and lowered into a preheated bath to 120 °C and stirred for 24 h. After the mixture was cooled, it was transferred to a separating funnel and organic products were extracted with 2M NaOH solution (3 × 10 mL). The combined organic extracts were washed with brine (3 × 15 mL), dried over Na_2_SO_4_, filtered, and evaporated *in vacuo*. The desired product was isolated by column chromatography.

##### 1,9-Diamino-*N*^9^-ethyl-*N*^1^-Octadecyl-*N*^3^,*N*^7^-dibenzyl-3,7-Diazanonane (**16a**)

Yield 410 mg (66%), colorless oil. Eluent: ACN-NH_3_·H_2_O (9:1). ^1^H NMR (400 MHz, CDCl_3_) δ 0.91 (t, *J* = 7.0 Hz, 3H, (CH_2_)_15_CH_3_), 1.07 (t, *J* = 7.1 Hz, 3H, CH_2_CH_3_), 1.29 (s, 30H, (CH_2_)_15_CH_3_), 1.44 (m, 2H, CH_2_CH_2_(CH_2_)_15_), 1.61–1.79 (m, 2H, PhCH_2_NCH_2_CH_2_CH_2_), 2.34–2.75 (m, 16H, 4 NHCH_2_, 4 NCH_2_), 3.55 (s, 4H, 2 PhCH_2_), 7.16–7.43 (m, 10H, 2 Ph). ^13^C NMR (101 MHz, CDCl_3_) δ 14.2, 15.1, 22.7, 24.8, 29.4, 29.7, 29.7, 29.8, 43.9, 47.2, 47.4, 49.9, 52.6, 52.7, 53.7, 59.0, 59.0, 126.9, 126.9, 128.2, 128.4, 128.8, 128.8, 134.2, 139.8, 139.8. MS ESI *m*/*z*: [M + H]^+^ calcd for C_41_H_73_N_4_ 621.58, found: 621.52.

##### *N*^1^-Ethyl-*N*^4^-[(*N*-Octadecyl)Aminoethyl]Piperazin (**16b**)

Yield: 31%, colorless oil. Eluent: ACN-NH_3_·H_2_O (95:5). ^1^H NMR (300 MHz, MeOD) δ 0.90 (t, *J* = 6.5 Hz, 3H, (CH_2_)_15_CH_3_), 1.10 (td, *J* = 3.2, 7.2 Hz, 3H, NCH_2_CH_3_), 1.31 (br.s, 30H, (CH_2_)_15_CH_3_), 1.55–1.75 (m, 2H, CH_2_CH_2_(CH_2_)_15_), 2.14 (td, *J* = 2.8, 12.5 Hz, 1H, NHCH_a_H_b_CH_2_N), 2.39–3.11 (m, 13H, Pip protons, NHCH_a_H_b_CH_2_N, NCH_2_CH_3_). ^13^C NMR (75 MHz, MeOD) δ 11.8, 14.5, 23.8, 27.1, 28.0, 30.5, 30.5, 30.7, 30.8, 33.1, 52.3, 52.5, 52.7, 53.2, 53.3, 53.7, 54.7, 56.5. HRMS ESI *m*/*z*: [M + H]^+^ calcd for C_26_H_56_N_3_ 410.4469, found: 410.4468. LCMS r/t: 11.78 min.

##### *N*^1^-[2-(Ethylamino)Ethyl]-*N*^4^-[*N*-(*rac*-1-Decyloxy-2-Ethyloxyprop-3-yl)Amino]Ethylpiperazin (**16c**)

Yield: 23%, colorless oil. Eluent: ACN-NH_3_·H_2_O (9:1). ^1^H NMR (600 MHz, CD_2_Cl_2_) δ 0.88 (t, *J* = 7.0 Hz, 3H, (CH_2_)_7_CH_3_), 1.03 (t, *J* = 7.2 Hz, 3H, NCH_2_CH_3_), 1.16 (t, *J* = 7.0 Hz, OCH_2_CH_3_, 3H), 1.29 (br.s, (CH_2_)_7_CH_3_, 14H), 1.51–1.57 (m, 3H, CH_2_CH_2_(CH_2_)_7_), 2.35 (q, *J* = 7.2 Hz, 2H, NHCH_2_CH_3_), 2.37–2.58 (m, 14H, 3CH_2_NH, Pip protons), 2.58–2.71 (m, 4H, CH_2_N(CH_2_CH_2_)_2_NCH_2_), 3.38–3.46 (m, 4H, CH_2_OCH_2_), 3.48–3.54 (m, 2H, CHOCH_a_H_b_CH_3_), 3.65 (dq, *J* = 7.0, 9.3 Hz, 1H, OCH_a_H_b_CH_3_). ^13^C NMR (151 MHz, CD_2_Cl_2_) δ 12.4, 14.3, 16.0, 23.1, 26.6, 29.8, 29.9, 30.0, 30.1, 30.2, 32.4, 47.1, 51.6, 52.6, 53.4, 53.8, 58.3, 65.6, 72.0, 72.3, 78.3. MS ESI *m*/*z*: [M + H]^+^ calcd for C_25_H_55_N_4_O_2_ 443.4, found 443.4. LCMS r/t: 8.31 min.

##### *N*^1^-Ethyl-*N*^4^-[*N*-(*rac*-1-Decyloxy-2-Ethyloxyprop-3-yl)Amino]Ethylpiperazin (**16d**)

Yield 184 mg (38%), colorless oil. Eluent: EA-MeOH-NH_3_·H_2_O (7:3:0.2). ^1^H NMR (300 MHz, CDCl_3_) δ 0.77–0.94 (t, 3H, (CH_2_)_7_CH_3_), 1.09–1.34 (m, 20H, (CH_2_)_7_CH_3_, OCH_2_CH_3_, NCH_2_CH_3_), 1.45–1.59 (m, 2H, CH_2_CH_2_(CH_2_)_15_), 2.31–2.82 (m, 16H, 2 NHCH_2_, Pip protons), 3.30–3.75 (m, 7H, CH_2_OCH_2_, CHOCH_2_). ^13^C NMR (75 MHz, CDCl_3_) δ 14.2, 14.2, 15.8, 22.7, 26.2, 29.4, 29.5, 29.6, 29.7, 29.7, 32.0, 43.8, 45.5, 46.5, 51.4, 53.2, 53.3, 53.6, 56.7, 57.4, 65.6, 71.7, 71.8, 77.4. HRMS ESI *m*/*z*: [M + 2Na]^2+^ calcd for C_23_H_49_N_3_O_2_Na_2_ 222.6805, found: 222.2215. MS ESI *m*/*z*: [M + H]^+^ calcd for C_23_H_50_N_3_O_2_ 400.4, found 400.4. LCMS r/t: 9.38 min.

##### *N*^1^-[2-[*N*-(Isopropylamino)ethyl]-*N*^4^-[*N*-(*rac*-1-Decyloxy-2-Ethyloxyprop-3-yl)Amino]Ethylpiperazin (**16e**)

Yield: 25%, colorless oil. Eluent: EA-MeOH-NH_3_·H_2_O (7:3:0.3). ^1^H NMR (300 MHz, CDCl3) δ 0.86 (t, *J* = 7.0 Hz, 3H, (CH_2_)_7_CH_3_), 1.08 (d, *J* = 6.3 Hz, 6H, CH(CH_3_)_2_), 1.18 (t, *J* = 7.0 Hz, 3H, OCH_2_CH_3_), 1.25 (d, *J* = 5.6 Hz, 14H, (CH_2_)_7_CH_3_), 1.47–1.60 (m, 2H, CH_2_CH_2_(CH_2_)_7_), 2.34–2.58 (m, 12H, 2 CH_2_N and Pip protons), 2.60–2.79 (m, 6H, 3 NHCH_2_), 2.82 (sept, *J* = 6.3 Hz, 1H, CH(CH_3_)_2_), 3.36–3.63 (m, 6H, CH_2_OCH_2_, CHOCH_a_H_b_), 3.69 (dq, *J* = 7.0, 9.3 Hz, 1H, CHOCH_a_H_b_). ^13^C NMR (75 MHz, CDCl_3_) δ 14.2, 15.9, 22.7, 22.8, 26.2, 29.4, 29.6, 29.7, 29.7, 29.8, 32.0, 43.8, 46.7, 49.2, 51.6, 53.4, 57.6, 57.7, 65.7, 71.8, 71.9, 77.7. HRMS ESI *m*/*z*: [M + H]^+^ calcd for C_26_H_57_N_4_O_2_ 457.4476, found: 457.4483. HRMS ESI *m*/*z*: [M + 2H]^2+^ calcd for C_26_H_58_N_4_O_2_ 229.2275, found: 229.2276. LCMS r/t: 8.39 min.

##### *N*^1^-[2-(N-Pentylamino)Ethyl]-*N*^4^-[*N*-(*rac*-1-Decyloxy-2-Ethyloxyprop-3-yl)Amino]Ethylpiperazin (**16f**)

Yield: 43%, colorless oil. Eluent: ACN-NH_3_·H_2_O (9:1). ^1^H NMR (300 MHz, CDCl_3_) δ 0.79–0.91 (m, 6H, (CH_2_)_7_CH_3_, (CH_2_)_2_CH_3_), 1.17 (t, *J* = 7.0 Hz, 3H, OCH_2_CH_3_), 1.25 (br.s, 18H, (CH_2_)_7_CH_3_, (CH_2_)_2_CH_3_), 1.41–1.58 (m, 4H, CH_2_CH_2_(CH_2_)_7_, NCH_2_CH_2_), 2.13–2.76 (m, 20H, 2 NCH_2_, 4 NHCH_2_, Pip protons), 3.32–3.60 (m, 1H, CH_2_OCH_2_, CHOCH_a_H_b_), 3.68 (dq, *J* = 7.1, 9.4 Hz, 1H, OCH_a_H_b_CH_3_). ^13^C NMR (101 MHz, CDCl_3_) δ 14.2, 14.2, 22.7, 22.8, 26.2, 29.4, 29.6, 29.6, 29.7, 29.7, 29.7, 29.8, 32.0, 46.6, 46.8, 50.1, 51.6, 53.4, 53.4, 58.0, 65.7, 71.8, 72.0, 77.8. HRMS ESI *m*/*z*: [M + 2Na]^+^ calcd for C_23_H_49_N_3_O_2_Na_2_ 222.6805, found: 222.2215. MS ESI calcd for C_28_H_61_N_4_O_2_ 485.5, found 485.5. LCMS r/t: 6.42 min.

##### *N*^1^-[(β-D-Glucopyranosyl)Aminocarbonyl]Methyl-*N*^4^-[(*N*-(*rac*-1-Decyloxy-2-Ethyloxyprop-3-yl)Aminocarbonyl]Methylpiperazin (**17**)

Yield: 89%, colorless oil. Eluent: EA-MeOH (8:2). ^1^H NMR (600 MHz, CDCl_3_) δ 0.87 (t, *J* = 7.0 Hz, 3H, (CH_2_)_7_CH_3_), 1.18 (t, *J* = 7.0 Hz, 3H, OCH_2_CH_3_), 1.20–1.33 (m, 14H, (CH_2_)_7_CH_3_), 1.50–1.58 (m, 2H, CH_2_CH_2_(CH_2_)_7_), 2.57 (br.s, 8H, Pip protons), 2.96–3.14 (m, 2H, 2 NCH_2_CO), 3.14–3.23 (m, 1H, H-6), 3.35–3.63 (m, 11H, H-1, H-2, H-3, H-4, H-6, CHOCH_a_H_b_CH_3_, CH_2_OCH_2_), 3.63–3.70 (m, 1H, OCH_a_H_b_CH_3_), 3.74–3.87 (m, 2H, CH_2_), 7.43 (br. t, *J* = 5.9 Hz, 1H, CH_2_NH), 7.99 (d, *J* = 8.3 Hz, 1H, CHNH). ^13^C NMR (151 MHz, CDCl_3_) δ 14.2, 15.9, 22.8, 26.2, 29.4, 29.6, 29.7, 29.7, 29.8, 32.0, 40.4, 40.5, 53.3, 53.6, 61.6, 65.5, 65.5, 71.4, 71.4, 72.0, 76.8, 76.9, 78.0, 79.8, 170.4, 171.9.

### 4.3. Cell Lines and Culture Conditions

All cell lines were obtained from N.N. Blokhin National Medical Research Center of Oncology cell collection. The following cell lines were used in the study: A-549 (lung carcinoma), MCF-7 (breast carcinoma), HCT116 (colorectal carcinoma), and HaCaT (human keratinocytes). Cells were cultured in Dulbecco modified Eagle’s medium (DMEM; PanEco, Moscow, Russia) with 10% fetal bovine serum (Biosera, France), mixture of the antibiotics penicillin and streptomycin in final concentrations of 50 I.U./mL and 50 µg/mL, respectively, and 2mM L-glutamine (both—PanEco, Moscow, Russia) in 5% CO_2_ at 37 °C.

Cells were seeded in 96-well plates (5 × 10^3^ cells/well) and treated with substances at concentrations from 200 to 1.5 µM for 72 h (5% CO_2_, 37 °C). Maximal DMSO concentration in the medium was 0.05%. Cell viability was determined using the MTT test as follows: cells were incubated for 4 h with 0.25 mg/mL solution of 3-(4,5-dimethylthiazol-2-yl)-2,5 diphenyltetrazolium bromide (MTT, D298931, Dia-M) (5% CO_2_, 37 °C). Following incubation, the medium was aspirated, and formazan was dissolved in DMSO (100 μL/well). The optical density of the solution was measured at 540 nm using a Multiskan Sky microplate spectrophotometer (Thermo Scientific, Waltham, MA, USA). The percentage of viable cells was calculated from the absorbance of vehicle control (0.5% DMSO). Each experiment was repeated three times, and each concentration was tested in three replicates (see Appendix A).

## Data Availability

Not applicable.

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
