# Peer review of "Synthesis of Novel Lipophilic Polyamines via Ugi Reaction and Evaluation of Their Anticancer Activity"

_molecules, 2022, doi:10.3390/molecules27196218_

Round 1
Reviewer 1 Report
The article by Nichugovskiy et al describes the synthesis of novel lipophilic polyamines employing a multicomponent reaction (MCR) and the results of the preliminary evaluation of their anticancer activity. The article could be published in Molecules, provided that minor revisions are applied.
Comments and corrections
11) Page 2, rows 56-59. One of the drawbacks in the synthetic approaches to polyamines (PA) is represented by the difficult procedures needed for the isolation and purification of these highly polar and soluble molecules. This should be mentioned in the list of drawbacks as lipophilic polyamines could be advantageously purified in distinct conditions.
2) Page 3, rows 87-89. The authors run the MCR in methanol at 65°C (reflux?) for 16 hours. The original protocol reports the N-split Ugi reaction running in methanol at room temperature. The authors should explain if heating is mandatory for a successful reaction with their specific substrates or not.
3) Page 4, row 115. The lower stability of the isocyanide 7c should be explained and supported.
4) Page 4, rows 137-138 and page 5, rows 157-159. The authors should report any experimental evidence of deacetylated derivatives supporting their claim for “partial deacetylation during reflux”.
5) Page 5, Scheme 4. The general procedure for the reduction is drawn as acting on N-methylamides and leading to N-methylamines. The scheme is misleading as neither compound 11c nor compounds 12b-d and 15a,b are N-methylamides. Please correct the Scheme accordingly (e.g.: RCONHR’ → RCH2NHR’).
6) Page 6, row 181. The authors claim excellent results of biological activity against tumor cell lines but they should specify that these results are preliminary.
7) Please check the whole paper for typos and grammar.
8) Materials and Methods. Please use 1 decimal figure for 13C-NMR chemical shifts.
9) Supporting information. Some NMR spectra show signals of related impurities and solvents, casting some doubts on the purity of the synthesized compounds. The authors should comment on this.
Author Response
Dear Reviewer,
Thank you very much for all your remarks.
According to your comments we revised our manuscript.
1) Page 2, rows 56-59. One of the drawbacks in the synthetic approaches to polyamines (PA) is represented by the difficult procedures needed for the isolation and purification of these highly polar and soluble molecules. This should be mentioned in the list of drawbacks as lipophilic polyamines could be advantageously purified in distinct conditions.
We totally agree with your comment and added this correction to the 59-60 line.
“Most of the known approaches regarding the synthesis of PA derivatives and their conjugates have several disadvantages [9], namely: 1) multistage synthetic procedures, 2) introduction of orthogonal protective groups to block internal and terminal nitrogen at-oms, 3) low overall yield of the desired molecules, and 4) complicated purification procedures that are required for highly polar compounds.”
2) Page 3, rows 87-89. The authors run the MCR in methanol at 65°C (reflux?) for 16 hours. The original protocol reports the N-split Ugi reaction running in methanol at room temperature. The authors should explain if heating is mandatory for a successful reaction with their specific substrates or not.
The N-split Ugi reaction could be proceed under room conditions. However, we used paraformaldehyde as the carbonyl component, which is a polymer with 8-100 monomeric units. To break paraformaldehyde it was necessary to reflux the reaction mixture. In addition, lipophilic isonitrile is insoluble in methanol at room temperature; heating is required to dissolve it completely.
“Usually the N-split Ugi reaction proceeds under room conditions, but we used refluxing to dissolve the lipophilic isocyanide and to break paraformaldehyde completely.”
3) Page 4, row 115. The lower stability of the isocyanide 7c should be explained and supported.
We found that isocyanide 7c was not formed by standard protocols from formamide (POCl3, Et3N or CBr4, Ph3P, Et3N or T3P®), but only using by treatment with I2, Ph3P under basic conditions. We detected 13C signals of formamide that was isolated from reaction mixture, which may indicate a low stability of isocyanide 7c.
The stability of isocyanides depends on their structure and substitution pattern. The lability of aliphatic decreases in the range tertiary > secondary > primary. [
https://doi.org/10.1002/ejoc.202101023; https://pubs.rsc.org/en/content/articlelanding/2020/gc/d0gc02722g; https://doi.org/10.1021/acs.chemrestox.9b00504; ].
For this reason, we obtained isonitrile 7c immediately prior to the reaction.
4) Page 4, rows 137-138 and page 5, rows 157-159. The authors should report any experimental evidence of deacetylated derivatives supporting their claim for “partial deacetylation during reflux”.
We suggested that a partial deacetylation of D-glucose occurred by treatment with basic piperazine. We observed a characteristic set of spots on TLC, the compound with the highest polarity corresponded to compound 17.
5) Page 5, Scheme 4. The general procedure for the reduction is drawn as acting on N-methylamides and leading to N-methylamines. The scheme is misleading as neither compound 11c nor compounds 12b-d and 15a,b are N-methylamides. Please correct the Scheme accordingly (e.g.: RCONHR’ → RCH2NHR’).
Corrected.
6) Page 6, row 181. The authors claim excellent results of biological activity against tumor cell lines but they should specify that these results are preliminary.
Corrected.
7) Please check the whole paper for typos and grammar.
Thank you for this comment, we have check for typos and grammar.
8) Materials and Methods. Please use 1 decimal figure for 13C-NMR chemical shifts.
Corrected.
9) Supporting information. Some NMR spectra show signals of related impurities and solvents, casting some doubts on the purity of the synthesized compounds. The authors should comment on this.
NMR spectra of compounds could contain residue signals solvent after chromatographic isolation, but before reactions each substance dried in a vacuum of for 2 h to remove any solvents. Before studying biological activity, each polyamine was also chromatographed and dried for 12 h under 4x10-3 torr. If we observed a significant amount of impurities, the substance was re-purified.
Sincerely,
Dr. Mikhail Maslov

Reviewer 2 Report
The article entitled Synthesis of novel lipophilic polyamines via Ugi reaction and evaluation of their anticancer activity presents the synthesis of new polyamines endowed with piperazine moiety. The rationale for designing these polyamines as anticancer is unclear and therefore needs to be implemented. The synthesis of new polyamines and their conjugates is well described and documented by the authors, also considering the spectra reported in SI. The experiments on the antitumor activity of some of the synthesized compounds are instead insufficient and consequently the title does not reflect the content of the article nor the assignment of this article in the special issue Anticancer Agents: Design, Synthesis and Evaluation III. However, with the necessary modifications and implementations, I suggest moving this contribution to the special issue of Molecules Polyamine Drug Discovery.
Author Response
Dear Reviewer,
Thank you very much for your comment.
1) The rationale for designing these polyamines as anticancer is unclear and therefore needs to be implemented.
Synthetic analogues of polyamines are anticancer agents (DOI: 10.1038/s41568-018-0050-3), some of which are in clinical trials. Studies of synthetic PA analogs show that they inhibit cell growth (DOI: 10.3389/fphar.2019.01670), activate apoptosis (DOI: 10.1016/j.bmc.2008.11.047). In addition, the lipophilic polyamine AMXT-1501, which contains palmitic acid, lysine and norspermine, effectively inhibits transport of PA (DOI: 10.1038/s41467-021-20896-z), and in the combination with DFMO increases cancer cells death. Also, bisalkylated derivative of norspermine (BENSpm) has passed Phase I and Phase II clinical trials and is considered a reference compound for anticancer polyamines in biological studies.
In the current paper, we demonstrated a new method for synthesis of novel lipophilic polyamines via the multicomponent Ugi reaction. For the first time it was shown that the presence of a diglyceride fragment in the structure of polyamines appears to be a key factor for the manifestation of high antitumor activity.
Recently we published the review in Molecules, SI Polyamines drug discovery concerning to the synthesis of polyamines and their conjugates, and their biological activity (DOI: 10.3390/molecules26216579), so we hope that our current manuscript is suitable for SI Anticancer Agents: Design, Synthesis and Evaluation III.
2) The experiments on the antitumor activity of some of the synthesized compounds are instead insufficient.
To evaluate the possibility of using the synthesized polyamines as antitumor agents, an initial cytotoxicity screening of the compounds was performed on several tumor cell models in vitro. We identified the structure-activity relationships and chose lead-compounds for further extended biological studies.
Sincerely,
Dr. Mikhail Maslov

Reviewer 3 Report
Reviewers’ comments for the Manuscript ID: Molecules-1888508
The manuscript title: “Synthesis of novel lipophilic polyamines via Ugi reaction and evaluations of their anticancer activity by Artemiy Nichugovskiy et al. Natural and synthetic poly amine are one of the interesting topics in cancer research. However, synthesis of PAs is complicated process because of different physicochemical properties of poly amines, in the current manuscript authors describes the method for the synthesis of several new LPAs comprising alkyl diglyceride with piperazine fragment via the multicomponent Ugi reaction and subsequent steps. All the synthesized compounds were thoroughly characterized by using different analytical method. It is interesting that, some of the newly synthesized molecules were exhibited comparable antitumor activity with know bisalkylated norspermine 3 (BENSpm) and Cisplatin besides. It is well written and organized manuscript, so I would strongly recommend publication of this article in molecule journal in its current formate.
1) Authors can find few more general comments in attached manuscript
2) Authors provided the NMR spectral details of all synthesized molecule in both manuscripts and supporting information file over spectra, so spectral details in supporting file may not require.

Author Response
Dear Reviewer,
Thank you very much for your comments.
1) Authors can find few more general comments in attached manuscript
In accordance with your comments, we have revised our manuscript and corrected the comments you pointed out in the article.
2) Authors provided the NMR spectral details of all synthesized molecule in both manuscripts and supporting information file over spectra, so spectral details in supporting file may not require.
Corrected.
Sincerely,
Dr. Mikhail Maslov
Round 2
Reviewer 2 Report
The author did not modify the paper according to the referee suggestions.
It is well Known that PA are studied as anticancer agents and and the authors were not required to write a lecture on DFMO and PA as an explanation of their work to the reviewer, but they must explain the rationale for using their polyamines as anticancer and support it with more robust data.
The experiments on the antitumor activity of some of the synthesized compounds are insufficient and the paper. The article cannot be published in this form and even more so for a special issue Anticancer Agents.
Author Response
Dear Reviewer,
Thanks a lot for your critical evaluation of the manuscript. We apologize for misunderstanding that arose during the previous review step and modification of the manuscript. Here we address all critical points in the revised manuscript version in a point-by-point basis. To make it easier for the reviewer to find the textual alterations, we submit a marked version of the manuscript as supplemental file, in which the major alterations are clearly labeled.
The proposed structure of lipophilic PAs comprises three major domains: the lipophilic part, the PA domain, and the short-chain alkyl substituent.
1) The lipophilic PA may effectively inhibit PA transport into the cell due to its effective incorporation into the transmembrane channel located on the cell membrane. These data have been previously reported for AMXT-1501 with the palmitic acid residue (DOI: 10.1038/s41467-021-20896-z). In addition, we have previously shown that lipophilic PAs, where the lipophilic part is presented by a diglyceride fragment, also exhibit high antitumor activity (DOI: 10.1016/j.mencom.2019.11.003). Considering the results of the mentioned above studies, conjugation of PAs with the diglyceride fragment may have beneficial pharmacological potential.
This issue has been addressed at Page 2 (lines 72-78) of the revised version of the manuscript.
2) The replacement of aliphatic diamines used on the initial stages of our synthetic strategy with piperazine results in increased conformational rigidity and lipophilicity (DOI: 10.1016/j.tet.2009.08.034), alteres proteolytic and, thus, may alter biological activity of resulting PAs. Previous works have demonstrated a high antiproliferative activity of various piperazine derivatives against different cancer cell lines (PMID: 23898056).
This issue has been already addressed at page 3 (lines 112-116) of the original manuscript version.
3) Finally, “the presence of an alkyl group at the terminal nitrogen atom of polyamine moiety slightly increases cytotoxicity compared to non-alkylated analogues. This effect may be related to the fact that the terminal alkyl group prevents potential acylation and further oxidation of the compound, which increases its stability in cells.”
This issue has been addressed at Page 2 lines 64-68 of the revised version of the manuscript.
4) In this work we used a standard protocol to test preliminary antitumor activity of new lipophilic PAs using MTT-test. The results already obtained allow us to make an unambiguous conclusion about the presence of antitumor activity. Undoubtedly, the use of novel compounds in a practical way will require more extensive and thorough studies, but they are well beyond the scope of this work, which will be carried out as a follow-up study.
We addressed this issue on page 6, lines 204-205.
We hope that we have adequately addressed the critical points of the reviewer and that the revised version of the manuscript may now be acceptable for publication.
Sincerely,
Prof. Mikhail Maslov